# Adaptive Autonomous Protocol for Secured Remote Healthcare Using Fully Homomorphic Encryption (AutoPro-RHC)

**DOI:** 10.3390/s23208504

**Published:** 2023-10-16

**Authors:** Ruey-Kai Sheu, Yuan-Cheng Lin, Mayuresh Sunil Pardeshi, Chin-Yin Huang, Kai-Chih Pai, Lun-Chi Chen, Chien-Chung Huang

**Affiliations:** 1Department of Computer Science, Tunghai University, Taichung 407224, Taiwan; rickysheu@thu.edu.tw (R.-K.S.);; 2Department of Industrial Engineering and Enterprise Information, Tunghai University, Taichung 407224, Taiwan; d08330003@thu.edu.tw (Y.-C.L.); huangcy@thu.edu.tw (C.-Y.H.); 3AI Center, Tunghai University, Taichung 407224, Taiwan; mayuresh@thu.edu.tw; 4College of Engineering, Tunghai University, Taichung 407224, Taiwan; lunchi@thu.edu.tw; 5Computer & Communication Center, Taichung Veterans General Hospital, Taichung 407224, Taiwan

**Keywords:** remote healthcare (RHC), federated protocol, fully-homomorphic encryption, cloud computing, heart diseases

## Abstract

The outreach of healthcare services is a challenge to remote areas with affected populations. Fortunately, remote health monitoring (RHM) has improved the hospital service quality and has proved its sustainable growth. However, the absence of security may breach the health insurance portability and accountability act (HIPAA), which has an exclusive set of rules for the privacy of medical data. Therefore, the goal of this work is to design and implement the adaptive Autonomous Protocol (AutoPro) on the patient’s **r**emote **h**ealth**c**are (RHC) monitoring data for the hospital using fully homomorphic encryption (FHE). The aim is to perform adaptive autonomous FHE computations on recent RHM data for providing health status reporting and maintaining the confidentiality of every patient. The autonomous protocol works independently within the group of prime hospital servers without the dependency on the third-party system. The adaptiveness of the protocol modes is based on the patient’s affected level of slight, medium, and severe cases. Related applications are given as glucose monitoring for diabetes, digital blood pressure for stroke, pulse oximeter for COVID-19, electrocardiogram (ECG) for cardiac arrest, etc. The design for this work consists of an autonomous protocol, hospital servers combining multiple prime/local hospitals, and an algorithm based on fast fully homomorphic encryption over the torus (TFHE) library with a ring-variant by the Gentry, Sahai, and Waters (GSW) scheme. The concrete-ML model used within this work is trained using an open heart disease dataset from the UCI machine learning repository. Preprocessing is performed to recover the lost and incomplete data in the dataset. The concrete-ML model is evaluated both on the workstation and cloud server. Also, the FHE protocol is implemented on the AWS cloud network with performance details. The advantages entail providing confidentiality to the patient’s data/report while saving the travel and waiting time for the hospital services. The patient’s data will be completely confidential and can receive emergency services immediately. The FHE results show that the highest accuracy is achieved by support vector classification (SVC) of 88% and linear regression (LR) of 86% with the area under curve (AUC) of 91% and 90%, respectively. Ultimately, the FHE-based protocol presents a novel system that is successfully demonstrated on the cloud network.

## 1. Introduction

The healthcare system is one of the high-priority factors for the country’s progress. Recently, the outburst of many contagious chronic diseases has highly affected major economies all around the world. Therefore, the use of IoT remote healthcare devices, also known as smart healthcare, has been beneficial to the population at all the age groups without a need to physically attend the hospital and visit emergency services [1]. 

However, the popularity of using various body-worn IoT devices has increased the risk of confidentiality and integrity loss which is risky for the social well-being of the patient [1,2]. To overcome such challenges, fully homomorphic encryption (FHE) is introduced in the medical healthcare context to provide secured healthcare services for improved healthcare service quality and contribution towards society [3]. The patient, as the end-user, will be freed from such an FHE system installation as it will be implemented on the middleware system/cloud. The purpose of FHE is to perform analytical functions over the encrypted data and provide solutions for the query request by the authorized user. The scope of this work is to ascertain the prediction of the critical condition or health issues within the remote patient equipped with healthcare sensors to be evaluated by the doctors in the hospital. The motivation is thought from “How can the affected remote patient’s data confidentiality be preserved for HIPAA compliance?” [4]. A similar problem is discussed by Sendhil R. and Amuthon A. for privacy preservation in the healthcare data exchange, which is further attempted to be implemented on the cloud but suffers major drawbacks [4,5,6]. Therefore, we have designed an FHE protocol that is demonstrated to be working using cloud computation as well as on a local workstation system. The recent FHE research works in the medical field are compared with our proposed AutoPro healthcare protocol, as given in the above Table 1. The FHE algorithms used in Table 1 comparison are different and include DCNN, K-means, Gorti’s/Carmicheal’s scheme, simple FHE, and TFHE. The detailed objectives set for this work before the design and implementation are given below. The health insurance portability and accountability act (HIPAA) is used to protect patient’s health data sensitivity and consent-based disclosure. The HIPAA privacy rule is implemented as the protected health information (PHI) and the security rule is its subset of information. Communication with the patient while information transmission consists of a subset of protected information or electronic-PHI (e-PHI).

The HIPAA security rule covers (a) the information by ensuring confidentiality, integrity, and availability; (b) knowledge about securing the e-PHI data; (c) avoidance of impermissible use; and (d) workforce compliance co-operation. Therefore, AutoPro-RHC helps to serve by ensuring the security rule of HIPAA compliance. The AutoPro-RHC objective of designing a novel secure communication protocol using a federated system, which processes only FHE data, is aligned with HIPAA compliance.

### 1.1. Objectives

Construction of a pre-processing algorithm for the open dataset.

Use of an open dataset shows the adaptability of the designed algorithm to the international standards. Applying a pre-processing algorithm can overcome the missing data and incomplete data problems. The selection of necessary features is important to provide efficient results and reduce the overhead on the computational costs. The FHE algorithm is designed to process the encrypted data and evaluate the necessary features as per the medical examiner’s requirements. FHE is known for preserving the confidentiality and integrity of encrypted data, thus making it an ethical evaluator with the HIPAA compliance and GDPR regulations [1]. Nevertheless, the designed system must be efficient on the target platform for the lightweight scheme to be utilized;

2.Design of a novel secure communication protocol for autonomous servers.

The problem of coordination amongst different prime and local hospital servers is solved by the adaptive autonomous protocol. The AutoPro is designed to effectively coordinate between prime hospitals to execute FHE computations by utilizing the local hospital’s patient data from the group of authorized hospitals. The test set of patients is taken in the encrypted form and the results are communicated in return after the approval by the respective hospital’s medical examiner. The model is evaluated on both platforms of the workstation as well as on cloud instances as a multi-party communication protocol. Therefore, the private dataset and FHE models of the prime hospitals are confidential within the network.

3.To demonstrate the working of AutoPro-RHC protocol in the cloud computing.

The AutoPro eliminates the dependency on the third-party systems of an authentication server (AS) and ticket-granting server (TGS). Also, the FHE system is designed to provide homomorphic machine learning-based evaluation with the patient’s data end-to-end encrypted; no private key sharing is required within this protocol. Successively, AutoPro job scheduler dynamically adapts to the severity of the patient conditions in the AWS cloud. The flowchart given provides a stepwise process performed by this protocol in the cloud. The protocol steps are also given exclusively for the detailed analysis. Whereas, the two algorithms provide the federated system operations and utilize FHE functions for the remote healthcare protocol.

4.Evaluation of the performance of the multiple datasets and different FHE-ML algorithms.

AutoPro-RHC experiments use three different open heart disease datasets as Cleveland, Hungary, and V.A. as well as four FHE-ML algorithms as linear regression, support vector machines (SVM), eXtreme gradient boosting (XGB), and a decision tree. In the beginning, three different datasets are used for training on three different prime servers. Successively, the patient’s record is evaluated by FHE-ML algorithm-based prime servers which are later shared with the medical examiner for approval. Ultimately, the implementation of machine learning and FHE-ML on workstations and the cloud for comparison provides a complete evaluation analysis. Thus, to determine optimal performance, the AutoPro-RHC protocol is evaluated on different open datasets and demonstrated.

### 1.2. Applications

Remote patients constitute a major population outside hospital proximity with different categories of disease. Every patient is usually affected by a different category of disease. Patients affected with single or multiple diseases can be monitored regularly for checkups with the respective sensor devices. Therefore, as a mobile device with a wrist sensor is used in this work for FHE-based heart disease evaluation, the following similar fashion devices can be utilized for the respective diseases:Glucometer for Diabetes Monitoring: One of the chronic health conditions known as diabetes is measured by excessive sugar in the blood due to incapability to produce insulin in the human body. As the blood sugar is converted into energy by insulin, absence of it may lead to severe health problems such as vision loss and heart and kidney diseases. The glucometer is used to test the blood sugar level and report the reading to the remote hospital;Blood Pressure Cuff: A prominent method to check a patient’s health is by heart rate and blood flow for the blood pressure checkup. The artery motions are transmitted in real time by the blood pressure cuff which indicate the possibility of hypertension, heart failure, kidney problems, and diabetes. The blood pressure cuff is applied on the upper arm to measure for pressure monitoring;Pulse Oximeter for COVID-19: The pulse oximeter is a multi-purpose device that can measure low blood oxygen level (SpO_2_), lung functioning, and heart rate in bar graph form. The chronic conditions for heart/lung issues includes pneumonia, asthma, and COVID-19 monitoring. This device is easily attached to the patient’s finger as a non-invasive clip;Electrocardiogram (ECG) + Stethoscope for Cardiac Conditions: The heart functions are captured by ECG whereas the heart, lung, and bowel sounds are captured by stethoscope. The occurrence of cardiac conditions includes arrhythmias or coronary artery disease. This device is placed on the patient’s chest to virtually monitor heart and lung sounds for cardiac assessment.

### 1.3. Literature Survey

An IoT healthcare device used for monitoring patients’ health is secured by somewhat homomorphic encryption, as presented by V. Subramaniyaswamy et al. [11]. A smartwatch captures the user’s health information which is encrypted with a dynamic key and then permuted data are processed with block encryption as a homomorphic function before storing on the cloud and later evaluated for performance. A scalable homomorphic encryption algorithm for cancer-type prediction is demonstrated by E. Sarkar et al. [12]. The genetic information dataset is used to predict cancer type from a novel logistic regression model using fast matrix multiplication for high dimensional features. Inference to encrypted data by privacy preserving is presented by S. Sayyad et al. [13]. This work uses a simple neural network with the HELib algorithm on MNIST and heart disease datasets for evaluation. The privacy-preserving CNN models with BFV homomorphic encryption are demonstrated by F. Wibawa et al. [14]. A secure multi-party protocol is used for deep learning model protection at each client/hospital, which collaborates by federated learning and evaluates by aggregating servers. A medical resource-constrained mobile device used for private decision tree-based disease detection is presented by S. Alex et al. [15]. An energy-efficient FHE compatible rivest scheme is proposed that works within the user edge device and in the cloud for the homomorphic operations. A secure two-party computation for cancer prediction using homomorphic encryption is demonstrated by Y. Son et al. [16]. A gated recurrent unit (GRU) model is used to secure and compute over the encrypted data for homomorphic encryption to predict end-to-end recurrence. Privacy preservation for precision medicine using machine learning is presented by W. Briguglio et al. [17]. A machine learning encryption framework is proposed to work between client-server models with genomics datasets. A multiple-feature pre-processing classifier is used with three different HE-compatible classifiers. A genetic algorithm (GA) for augmented ensemble learning for FHE is demonstrated by D. Dumbere et al. [18]. The designed model consists of configuration settings, the evaluation of best configuration by GA, a training set for machine learning, a classifier pool of different CNN models, instance matching, and FHE evaluation for encrypted email spam filtering. Detection of COVID-19 by FHE in the federated system is presented by F. Wibawa et al. [19]. A secure multi-party computation protocol is used in a centralized federated system for protecting the aggregation of an encrypted CNN model weight matrix for the patient’s personal medical data. 

Sharing informatics for integrating biology and the bedside (I2B2) aggregate medical data, secured by FHE in the cloud, is demonstrated by J. Raisaro et al. [20]. The model consists of a public key encrypted I2B2 dataset stored on a cloud server in encrypted form and a shared key in the proxy server with an exclusive crypto engine and interacting with the client app. The database extract, transform, and loading (ETL) concepts are used for interactions in this process. An FHE-secured query system for a medicine side effect search is presented by Y. Jiang et al. [21]. In case of an un-trustable cloud server, a client-side server as middleware is added which keeps the public keys that take queries from the client terminal, encrypts it, and performs transactions on the cloud database server. The server helps to perform the medicine side effect search using FHE. Secure medical cloud computing using FHE is demonstrated by O. Kocabas et al. [22]. The patient’s cardiac health medical data for long QT syndrome (LQTS) detection is stored on the cloud in encrypted form and is evaluated by a medical examiner with HElib for the purpose of remote health monitoring. Securing deep neural network models by privacy-preserving machine learning (PPML) with FHE is presented by JW Lee et al. [23]. A ResNet-20 model with RNS-CKKS in FHE is implemented with bootstrapping on the CIFAR-10 dataset with approximate methods to evaluate non-arithmetic functions used with ReLU and softmax. Securing deep convolutional networks (DCN) with FHE is demonstrated by S. Meftah et al. [24]. A new DCN model with low depth and batched neurons is designed to be utilized by FHE for better performance. Multi-party computations by machine learning for MNIST data analyzed with privacy preserving is presented by T. Li et al. [25]. A non-interactive system with a multi-layer perceptron model and security protocols are presented with secure multi-party schemes to reduce calculations. A multi-key HE (MKHE) system for detecting disease-causing genes is demonstrated by T. Zhou et al. [26]. MKHE is combined with an encrypted pathogenic gene location function for operating on two location circuits, namely threshold (TH)-intersection and top gene position, as fixed parameters (Top-q) to locate polygenic diseases. Federated learning-based PPML with FHE is presented by H. Fang et al. [27]. The encrypted gradients are passed by the multiple parties to be combined In the federated learning model with partial homomorphic encryption. Thus, a federated MLP is implemented to compute backpropagation with gradient exchange. Federated analytics for multiparty homomorphic encryption (FAMHE) in precision medicine is demonstrated by D. Froelicher et al. [28]. FAMHE secures distributed datasets while including Kaplan-Meier survival analysis and medical genome evaluation. FHE, with full domain functional bootstrapping, is presented by K. Klucznaik et al. [29]. A regev encryption is used to compute affine functions which drastically reduces errors and performs FHE additions and scalar multiplications with high efficiency. Securing enterprises with a multi-tenancy environment is demonstrated by P. Dhiman et al. [30]. This work presents an enhanced homomorphic cryptosystem (EHC) which works with the BGV scheme for key and token generation and is implemented in an enterprise environment having a token, private, hybrid, and public-key environment. This FHE literature survey analysis is shown in the form of the abstract as the Figure 1 for different homomorphic encryption types, AI algorithm integration, communication protocols used, key distribution strategies, and applications.

### 1.4. The Survey Limitations Are Given as Follows

The complete mathematical model is not represented in most of the recent work;The FHE algorithm and libraries used are not specified in the implementation section;The details of every FHE algorithm hyper-parameter tuning are not disclosed;Comparison of machine learning and concrete-ML experiments with different open datasets and algorithms used in the protocol.

## 2. Materials and Methods

Figure 2 presents the federated model designed for the implementation of secured remote monitoring (SRM). The federated model can be explained in four parts as cloud learning, prime institutions with private data training, child institutions for test set submission, and user’s consisting of the patients and medical examiners. At the start, the patient shares their encrypted remote sensor’s data with the AutoPro network. Next, in the AutoPro network at layer 1, the hospital’s cloud portal with storage and computation is present that is used to collect queries as prime and from local hospital servers. Successively, the queries are resolved with the support from prime hospital servers with results and respond back to the requesting hospital by the prime server. The prime hospital servers consist of their private servers with the FHE algorithm.

The FHE algorithm is trained on the private dataset of that respective national/international hospital on a regular interval, i.e., weekly/monthly, and then serves for responding to queries. In layer 2, local hospitals can only submit queries to receive the results from the expert’s opinion formed collectively. Here, the dataset is not standardized due to the lack of policy and standardization procedures. Ultimately, the output forwarded by the source prime/local hospital to the respective medical examiner can receive the patient’s current health conditions and can share the approved medical report with them.

Figure 3 presents the flowchart for the AutoPro-RHC process. At the start, the sensor device initiates the medical reporting of a remote patient at a given interval of time. If the data are corrupted, the process is then terminated. Otherwise, the remote patient’s data are encrypted and sent to the respective local/prime hospital where the remote patient is registered. If the remote patient is registered at a local hospital, then their encrypted data are immediately forwarded to the federated cloud server. Otherwise, the prime hospital first sends the remote patient’s encrypted data to the cloud and starts FHE computation based on the training of the current prime hospital’s data. If the cloud service queue is full due to many requests of compute intensive FHE computations, then the process has to wait until the service queue slot availability. Next, the cloud will send a copy of the patient’s encrypted data to all prime hospital servers to perform FHE computation and provide prediction based on their respective data. Later, the results are returned back by all the prime hospitals to the federated cloud server to perform grouping of the results, which is then returned back to the requesting patient. The patient will later consent and share the results with the hospital’s medical examiner who will approve and return the final report with the patient.

Figure 4 the AutoPro job scheduler decides the scheduling of the patient’s data evaluation based on the recent heart conditions. The mode value is selected by first obtaining the highest value from the TestSet data submission of the patient and then, based on age, the heart rate can be selected as severe, moderate, and slightly affected for the modes 1, 2, and 3, respectively. Subsequently, in the processed version of the UCI dataset, the disease presence is already given in the feature column of the severity level for existing heart disease.

In case of Mode = 1, the critical care is reserved for the patients with severe heart conditions. Thus, severe heart conditions can increase the mode value to 1 and can schedule the results evaluation time to the hospitals alert/notice. Therefore, the scheduler used by the AutoPro job network is for parallel processing by all of the prime hospital servers, as shown in Figure 4a. Successively, the Mode = 2 with special care is reserved for the patients with mildly affected heart conditions. Thus, the scheduler used by the AutoPro network is the relay based scheduler, as shown in Figure 4b, which forwards the patients’ data as TestSet with results to the successive hospital prime server. Similarly, the Mode-3 with general care is allotted for the patient’s with slightly affected heart conditions. The scheduler used by the AutoPro network is the buffer based time synchronization where a buffer with less than or equal to 10 patients TestSet [10] is forwarded every 1~5 min to the successive server for processing, as shown in Figure 4c. Ultimately, all the results are gathered back to the initiating prime server.

### 2.1. Protocol and Flowchart Design

The FHE protocol process can be given as communication between the remote patient, federated network, and the medical examiner interaction process:
The remote patient initiates the process by sending encrypted sensor data at time T to the respective prime/local hospital with the user’s public key ${PK}_{1}$. The prerequisite is that the patient needs to be registered within the respective hospital for the treatment;
$$RemotePatient\;\to \;HospitalServer({Enc}_{\begin{array}{c} PK1 \end{array}}\left(SensorsData\left[T\right]\right))$$The prime/local hospital uploads the encrypted data to the federated cloud server for processing by Algorithm 1 in the federated cloud. The modes for severe, mild, and slightly infected are given as Mode 1, 2, and 3, respectively. In the case of slightly infected cases, a batch of jobs are transferred in the form of relay between the servers as TestSet[n];
HospitalServer → FederatedSystem(TestSet[])


If (TestSet(Mode) == 1)
$$PrimeServer1.FHE\left(TestSet,1\right)\;\to \;PrimeServer2.FHE\left(TestSet,1\right)\;\;\&\&\;PrimeServer3.FHE\left(TestSet,1\right)$$

ElseIf (TestSet(Mode) == 2)
$$PrimeServer1.FHE\left(TestSet,2\right)\;\to \;PrimeServer2.FHE\left(TestSet,2\right)\;PrimeServer2.FHE\left(TestSet,2\right)\;\to \;PrimeServer3.FHE\left(TestSet,2\right)$$

ElseIf (TestSet(Mode) == 3)
$$\begin{array}{l} PrimeServer1.FHE\left(TestSet[n],3\right)\;\to \;PrimeServer2.FHE\left(TestSet\left[n\right],3\right)\\ PrimeServer2.FHE\left(TestSet\left[n\right],3,Results2\right)\\ \;\to \;PrimeServer3.FHE\left(TestSet\left[n\right],3,\;Result2,\;Result3\right) \end{array}$$

Else
(“Invalid Request”)
3.The AutoPro then shares the patient’s encrypted data copy for evaluation to the respective trained FHE server model given in Algorithm 2. The encrypted results are then grouped by the receiving prime server and communicated back to the patient;
$$ResultSet=\sum_{0}^{N}PrimeServer.FHE\left(TestSet\left[N\right]\right)\;$$
RemotePatient← ResultSet

4.The patient decrypts the results with his private key ${SK}_{1}$ and can inspect the confidential results;
$$RemotePatient\;\leftarrow \;Results={Dec}_{{SK}_{1}}\left(ResultSet\right)$$
5.If the remote patient forgets/delays to share results with the medical examiner, then they receives a reminder from the hospital server. The hospital prime/local server stores the patient’s history and progress at regular intervals to remind them of the test reports;$$\begin{array}{l} \text{HospitalReminder}()\\ \text{If}(\text{CurrentDate}==\text{StartofWeek}())\\ \;\mathrm{For}(\forall \;PendingPatients)\\ \;\text{SendReminder(“Your results evaluation is pending.”)}\\ \;if\left(HospitalReminder\right)\\ \;Alert\left(”Your\;results\;evaluation\;is\;pending.”\right) \end{array}$$
6.The patients who are keen to evaluate the report by the medical examiner with/without a reminder from the hospital send the message with their public key for identification and then encrypt the results with the hospital public key ${PK}_{2}$ and forward it;
$$RemotePatient\;\to \;ProcessResults({PK}_{1},{Enc}_{{PK}_{2}}\left(Results\right))$$
7.The medical examiner then evaluates the report by decryption using his private key ${SK}_{2}$, averages the results, and provides further prescription on the current patient’s status;
$$\begin{array}{l} \mathrm{ProcessResults}(PublicKey\;{PK}_{1},String\;{Enc}_{{PK}_{2}}\left(Results\right))\\ \;\mathrm{MedicalExaminer}\leftarrow {Dec}_{{SK}_{2}}(Results)\\ \;\mathrm{EvaluatedReport}={MedicalExaminer}_{Eval}(Results)\\ \;\mathrm{Report}={Enc}_{{PK}_{1}}(EvaluatedReport)\\ \text{Return Report} \end{array}$$
$$\;\;\;\;\;\;MedicalExaminer\to \;EvaluationReport={Dec}_{{SK}_{2}}{(Enc}_{{PK}_{2}}\left(Results\right))$$
8.The evaluation is then communicated back to the patient by encryption with their public key ${PK}_{1}$;
$$RemotePatient\leftarrow {Enc}_{{PK}_{1}}(EvaluationReport)$$
9.Therefore, the patient is kept informed securely about their health status with the report decrypted by ${SK}_{1}$.
$$RemotePatient\leftarrow {Dec}_{{SK}_{1}}\left({Enc}_{{PK}_{1}}(EvaluationReport)\right)$$


### 2.2. Algorithm Description

Algorithm 1 presents the process for the federated system. This algorithm is executed on the cloud server to provide services to the patients and medical examiners using AutoPro-RHC. In step 1, the input taken is FederatedQueue which determines the size of the cloud server queue assigned to every local/prime hospital for job requests. In step 2, the PrimeServer specifies the group of prime server addresses {PS1, PS2, …, PSN} that are needed to execute the FHE algorithm on their private dataset. In step 3, the LocalServer is the group of approved hospital servers that can make requests for the TestSet evaluation by the cloud server. In step 4, the AuthorizedServer is given as all the servers local/prime involved within this model. In step 5, TestSet[n] is the group of test requests sent by an authorized hospital’s server. In step 6, the output given by this algorithm is the ResultSet[] which is returned as the successful processing of the job request by the authorized servers. In step 7, all the variables used within this algorithm are initialized to NULL. In step 8, the global variable is declared as NULL, which can be used consistently across the code structure. In step 9, the event needs to assign a value that can be either 0 for weekly training or 1 for monthly training of the prime servers for the FHE evaluation with recent training data. In step 10, the if condition checks whether the current day is equal to the event day. In step 11, if the condition is true then the signal variable is compared with the prime server group to determine whether all the prime servers are trained for this event. In step 12, if the previous condition is true then the message is printed as prime servers are updated with training. In step 13, if the prime systems are considered to be in the training stage then are displayed with a message given in step 14. In step 15, all the prime servers are instructed for recent training by the remote procedure call which is counted as true (1) in step 16 for every successful signal value. In step 17, the TestSet[] is received from all the authorized servers requesting federated cloud service. In step 18, the condition checks whether the TestSet[] has requests pending and then in step 19, whether the FederatedQueue size allotted to that particular authorized server is full or not is checked. In step 20, if the condition is true then whether the TestSet[] belongs to the prime server’s request is checked and then, except for that prime server, the request is forwarded to other prime server as it is already processing it. Next, the results of the other prime server’s FHE computation are processed in step 21. Whereas, the results of the requesting prime server are added after availability in step 22. In step 23, the TestSet[] is forwarded to all the prime servers for FHE computation and stored in the ResultSet[] in step 24. In step 25, if the federated queue is full for that respective prime server then it is displayed for the request stage in the wait queue of the cloud server in step 26. Ultimately, in step 27, the TestSet[] is returned to the respective authorized servers.
**Algorithm 1:** AutoPro Job Distribution Algorithm**Input**: *FederatedQueue*, Size of federated queue on cloud server.   *PrimeServer*, Physical address of each prime server             {*PS*_1_, *PS*_2_ …, *PS_N_*}.3.   *LocalServer*, Physical address of each local server              {*LS*_1_, *LS*_2_, …, *LS_N_*}.4.   *AuthorizedServer*, ∀ (PrimeServers ⋃ LocalServer).5.   *TestSet*[], A group of recent remote health patients data sent by            authorized hospital’s server.6.**Output**: *ResultSet*[], Prediction for the *TestSet*[] send by the                  *AuthorizedServer*.7.Initialize (*FederatedQueue*, *PS*_1_, *PS*_2_, …, *PS_N_*, *LS*_1_, *LS*_2_, …, *LS_N_*,*       TestSet[], Signal*) *=* ∅8.Global *ResultSet[]=*∅9.*Event = Value   #* Assign 0 = Start of Week or 1 = Start of month.10If (*CurrentDate* == Event)11.   If Size(Signal) ==Size(*PrimeServer*)12.      Print “All prime servers are in ready state”13.   Else14.      Print “Prime server is still under training”       # Wait until ∀ *PrimeServer’s* are trained.15.      For ∀ *PrimeServer*16.     Signal + = RPC (Train *PrimeServer*)          # Training with updated data.17.*TestSet*[]=JobRequest(*AuthorizedServers*)18.If(TestSet[]!=NULL)19.  If(∀ *PrimeServer*Queue!=FULL)20.    If TestSet[] ϵ *PrimeServer*21.      ResultSet[] = $\sum_{0}^{N-1}PrimeServer.FHE\left(TestSet[]\right)$22.        ResultSet[] += ${PrimeServer}_{i}.Result$
23.    Else24.         Wait(QueueAvailability)25.         ResultSet[] = $\sum_{0}^{N}PrimeServer.FHE\left(TestSet[]\right)$26.  Else27.    Print “Request in wait queue of the cloud server”28.Return *ResultSet[]*

The Algorithm 2 for RHM-FHE consists of the following steps: Step 1 is an input of RemoteDataset[] as the open dataset used for training the FHE algorithm with heart disease. In step 2, Labels[] are the shortlisted features that will be used for the FHE algorithm as an input. In step 3, the TestSet[] is the recently collected remote patient’s health data for the purpose of health status evaluation. In step 4, the output given by this algorithm is candidateSet[], which is the prediction results by different FHE algorithms using a decision tree, logistic regression, SVM, and XGBoost. In step 5, the FHE algorithm time specifies the execution time required for the algorithm on the respective platform. In step 6, all the variables are initialized to NULL. In step 7, the remote dataset is divided into two parts of DTrain and DTest by an 80-20 ratio with the selected labels, respectively. In steps 8–9, the DTrain and DTest are normalized with the min–max normalization method for preprocessing of data within the range [0,1], respectively. In step 10, the FHE algorithm is trained with *DProcessed_Train_* and *DProcessed_Test_* by using a fast fully homomorphic encryption over the torus (TFHE) library with a ring-variant of the GSW [31,32]. In step 11, the DQueue stores the 10 patient’s test data TestSet received from the cloud server. Next, in step 12, if the DQueue data received are consistent then a message is printed that the received data are consistent and ready to be processed in step 13. In step 14, due to the received corrupted data, respective message is printed and the algorithm is terminated by the exit function in step 15. In step 16, the for loop is initiated to loop until the size of the TestSet patient’s data received. In steps 17–20, the candidateSet1, candidateSet2, candidateSet3, and candidate Set4 stores the FHE health status prediction results by the FHE algorithms of concrete-ML by decision tree, logistic regression, SVM, and XGBoost for the 10 patient’s remote data, respectively. Therefore, in step 21, the prediction results received from the previous candidateSet’s for different FHE algorithms are printed. In step 22, the time required for all four FHE algorithms is printed for analyzing the execution time. In step 23, all the candidate sets are combined to be stored in the candidate set array. Finally, in step 24, the algorithm returns the candidateSet[] to the calling cloud FHE function.
**Algorithm 2:** Adaptive AutoPro for Healthcare Prediction using the Fully Homomorphic Encryption (FHE) Algorithm**Input**: *RemoteDataset[]*, Open dataset of remote users for the training of heart disease.    *Labels[]*, Selected dataset labels to be processed by FHE algorithm.    *TestSet[]*, A group of recent remote health patients’ data sent by          An authorized hospital’s server.4.**Output**: candidateSet[], Prediction given by FHE algorithm for the          patient’s health status.5.    *FHE_AlgorithmTime[]*, Time required for the execution of the FHE algorithm. 6.Initialize (*DTrain*, *DTest*, *DProcessed_Train_*, *DProcessed_Test_*, FHE Model, DQue, candidateSet1, candidateSet2, candidateSet3, candidateSet4) = ∅7.DTrain, DTest=*TrainTestRatio*(RemoteDataset[], Labels, 80:20)8.${DProcessed}_{Train}=\frac{DTrain-Min\left(DTrain\right)}{Max\left(DTrain\right)-Min\left(DTrain\right)}\;$# Min–Max normalization by rescaling the data [0,1].9.${DProcessed}_{Test}=\frac{DTest-Min\left(DTrain\right)}{Max\left(DTrain\right)-Min(DTrain)}$10.FHE = FHE.Train(*DProcessed_Train_*, *DProcessed_Test_*)# TFHE library with GSW11.DQueue = Input (TestSet[10]) # Input encrypted Test Set of 10 patients.12.If (*DQueue* == Consistent)13.  Print “The patient’s test set is consistent and ready to be processed”14.Else15.  Print “The patient’s test set is corrupted”, exit() 16.For EHR in range(sizeof(TestSet[]))17.  candidateSet1 = FHE.Decision Tree(TestSet[x])18.  candidateSet2 = FHE.Logistic Regression(TestSet[x])19.  candidateSet3 = FHE.SVM(TestSet[x])20.  candidateSet4 = FHE.XGBoost(TestSet[x])21.Print “Decision Tree:”,candidateSet 1,“Logistic Regression:”, candidateSet2, “SVM:”,candidateSet 3,“XGBoost:”,candidateSet 422.Print “Execution Time required for Decision Tree:”, candidateSet1.time(),“Logistic Regression:”,candidateSet2.time(), “SVM:”,candidateSet3.time(), “XGBoost:”,candidateSet4.time()23.candidateSet[] = candidateSet1 ∪ candidateSet2 ∪ candidateSet3 ∪ candidateSet424.Return candidateSet[]

### 2.3. Mathematical Model

The faster FHE (FFHE) improves the processing speed of the system model by using optimized FHE computations [32,33]. The bootstrapping key size is also reduced by using an approximation algorithm. During the FHE computations, some errors are generated known as learning with errors (LWE) whereas the ring variant is known as Ring-LWE. The LWE cipertexts with unified representation are TLWE encoding polynomials over the torus. The security of TLWE depends on ideal lattice reduction or general hardness.

For the lattice-based homomorphic encryption schemes, the construction of both LWE/Ring-LWE variants can be used. The torus contains the right number of the LWE sample and can also be described as a continuous Gaussian distribution. The scale invariant LWE (SILWE) is used to work on the real torus. TLWE samples are used as follows:

(a) Search Problem: There exists multiple random homogeneous TLWE samples which are polynomial and then find key $S\;\in \;B_{N}{\left[X\right]}^{K}$;

(b) Decision Problem: The difference between fresh random homogeneous TLWE samples and uniform random samples taken from $T_{N}{\left[X\right]}^{K+1}$ is distinguished. Where K ≥ 1 integer, TLWE secret key $S\;\in \;B_{N}{\left[X\right]}^{K}$ is K polynomial vector $\in R=\frac{Z\left[X\right]}{{\left[X\right]}^{K}+1}\;$having a binary co-efficient $T_{N}[X]$ in the TLWE sample message space.

TGSW can be defined as the FHE scheme’s GSW generalized scaled invariant version. The author’s gentry, sahai, and water proposed GSW with LWE problem-based security. The gadget decomposition function is utilized by this TGSW scheme for improving the processing time and minimal memory usage with small noise as an approximate decomposition function. The input given as a TLWE sample is $\left(a,b\right)=\left(a_{1},..\;a_{K},\;a_{K+1}\right)\in T_{N}{\left[X\right]}^{K}\times T_{N}[X]$ and p is an integer polynomial.
(1)$$\sum_{j=0}^{N-1}a_{i,j}X^{j}$$

Here, a unique representation is chosen for unique $a_{i}$ with $a_{i,j}\in T$ and set ${\bar{a}}_{i,j}$ as the nearest multiple of $\frac{1}{B_{g}^{l}}$ to $a_{i,j}$.
(2)$$\sum_{p=1}^{l}{\bar{a}}_{i,j}p\frac{1}{B_{g}^{l}}$$

The ${\bar{a}}_{i,j}$ is decomposed uniquely with each ${\bar{a}}_{i,j},p\;\in \left[-\frac{B_{g}}{2},\left.\frac{B_{g}}{2}\right]\right.$
(3)$$e_{i,p}=\sum_{j=0}^{N-1}{\bar{a}}_{i,j},pX^{J}\in R$$

Equation (3) is executed as a matrix with i=1 to k+1 and p=1 to l. The output returned is in the form of ${\left(e_{i,p}\right)}_{i,p}$ as a combination of
$$\left[e_{1,1},..e_{K+1,l}\right]\in R^{\left(K+1\right)l}\;\in \;{dec}_{i,j}=\sum_{p=1}^{l}\in i,p.\frac{1}{B_{g}^{p}}-a_{i,j}={\bar{a}}_{i,j}-a_{i,j}\;\leq \frac{1}{2B_{g}^{l}}=\in$$
as ${\bar{a}}_{i,j}$ is the closest multiple of $\frac{1}{B_{g}^{l}}$ on torus and is a concentrated distribution where $B=Set\left\{0,1\right\}.$

LWE Key-Switching Procedure:

LWE is given a sample of message μ∈T, the procedure of key switching KS with the same μ is output with less noise occurrence and tolerate approximations. The input given is the LWE sample $\left(a^{\prime }=\left(a_{1}^{\prime },\;..a_{n}^{\prime }\right),b^{\prime }\right)\;\in {LWE}_{S^{\prime }}(\mu )$ with key switching ${KS}_{a^{\prime }}\to a.$ Where $S^{\prime }\in \;{\left\{0,1\right\}}^{n^{\prime }},\;S\in {\left\{0,1\right\}}^{n}$ and t∈N is a precision parameter.
(4)$$|\bar{a_{i}^{\prime }}-a_{i}^{\prime }|<2^{-(t+1)}$$
where $\bar{a_{i}^{\prime }}$ is the nearest multiple of $\frac{1}{2^{t}}$ to $a_{i}^{\prime }$
(5)$$\bar{a_{i}^{\prime }}=\sum_{j=1}^{t}a_{i}^{\prime }.j.2^{-j}$$

$a_{i,j}^{\prime }\in \left\{0,1\right\}$ where each $\bar{a_{i}^{\prime }}$ is the binary decomposition.
(6)$$\left(a,b\right)=\left(0,b^{\prime }\right)-\sum_{i=1}^{n^{\prime }}\sum_{j=1}^{t}a_{i}^{\prime }j.{KS}_{i,j}$$

Equation (6) is the output LWE sample ${LWE}_{s}(\mu )$.

Bootstrapping Procedure: LWE sample ${LWE}_{s}\left(\mu \right)=(a,b)$ has an encryption of μ by key S as a bootstrapping construct with constant noise. The intermediate encryption scheme used is TLWE.
(7)$$ACC\leftarrow (X^{\bar{b}}.(o,\;testv)\in \pi_{N}{\left[X\right]}^{K+1}$$
where $\bar{\mu }=\frac{\mu_{1}+\mu_{0}}{2},{\bar{\mu }}^{\prime }=\mu_{0}-\bar{\mu },b=\left[2Nb\right]$ and ${\bar{a}}_{i}=\left[2Na_{i}\right]for\;each\;i\in [1,n]$.
(8)$$testv=\left(1+X+\cdots +X^{N-1}\right)\times X^{-\frac{2N}{4}}\mu^{-1}\in \pi_{N}\left[X\right].\;\;ACC\leftarrow [h+\left(X^{{-\bar{a}}_{i}}-1\right).{BK}_{i}]⊡ACC$$
(9)$$u=\left(0,\bar{\mu }\right)+sample\;extract(ACC)$$
where ${KeySwitch}_{K,S}(u)$ is the output. Here, the squashing technique utilized by the accumulator achieves an additional 2x speed up.

## 3. Results

In this section, the implementation details of the FHE-based RHM are given in detail. The system configuration required to implement the FHE operations on the workstation and cloud instance is given in Table 2 as follows:

Three independent datasets are referred for the FHE model training on the prime servers, as given in Table 2. The Table 3 dataset is referenced from the UCI Heart Disease Dataset (Cleveland, Hungary, and V.A.) [34]. The purpose of referring to the different open datasets shows the adaptability of the FHE model to international standards.

### 3.1. Pre-Processing Techniques Used on the Dataset

The pre-processing techniques applied on the different datasets, as shown in Table 4, are necessary to achieve the high accuracy for the model performance. Every dataset is pre-processed to handle missing data to maintain consistency. One-hot encoding deals with categorical data processing.

Therefore, string category data can be transformed into numbers for machine learning processing. The data imbalance is handled using class weights for adjusting cost function within the model classification for penalizing major/minority classes accordingly.

### 3.2. Machine Learning Performance for Different Algorithms on the Dataset (Train:Test Ratio = 80:20)

The above machine learning-based evaluation of the three different UCI datasets are presented in Table 5. Every prime server dataset is evaluated using multiple machine learning algorithms that include linear regression, support vector machines (SVM), XGBoost, and decision tree. The dataset for prime servers 1, 2, and 3 is trained with Cleveland, Hungary, and V.A. heart disease open datasets, respectively. A total of 10 parameters are used as input to the algorithms where the highest values in most of the metrics of precision, F1-score, and accuracy are recorded by the decision tree algorithm output with the minimum processing time overall. In the case of prime server 2, all the metrics perform highly for the SVM. Whereas, the lowest recorded time is for the decision tree across all of the prime servers due to its lowest algorithm time complexity O(Nkd) where n is the number of training data, k is the features, and d is the decision tree depth. The prime server 3 also performs better for SVM in most of the metrics with less algorithm execution time in comparison to other prime servers.

In the case of machine learning-based evaluation on the cloud server, it can be observed that the linear regression algorithm performs with less time in comparison to the other prime server algorithm implementations. Whereas, the SVM algorithm executed on different prime servers has a similar time execution as that of the workstation. The execution of the decision tree and XGBoost performed oppositely, having higher time requirements with the same process on the workstation server.

### 3.3. FHE Concrete-ML Performance

The FHE-based evaluation of the concrete-ML workstation is presented in the above Table 6. FHE computations take longer time in comparison to other cryptographic algorithms. FHE combined with machine learning is presented using concrete-ML operations. In Table 6, the dataset used for evaluation is similar to Table 4 prime servers. Therefore, all the FHE prime servers are evaluated based on similar machine-learning algorithms for comparison with this workstation. In prime servers 1, 2, and 3, the best scores achieved by the algorithms are by a decision tree, SVM, and SVM, respectively. It is found to be similar in the metric evaluation as compared to the workstation. Whereas, the major difference is noticed within the timing evaluation of the FHE computations on the workstation. Nevertheless, the FHE decision tree requires the lowest time in comparison to the other FHE-based machine learning algorithms. Implementing a concrete-ML algorithm on the workstation with the highest accuracy is observed on prime server 2 followed by prime server 1 and the lowest on prime server 3. In case of timing requirements, prime server 3 requires the lowest execution time followed by the primer server 2 and highest time for prime server 1. The confusion matrix for the concrete-ML based machine learning evaluations are presented in detail in Appendix A Figure A1, Figure A2, Figure A3 and Figure A4, respectively.

### 3.4. Details for the Parameters Tuned to Improve the Performance by the Concrete-ML Model

The support vector classification is based on libsvm with the configuration parameters as shown in Table 7. The number of samples is limited to thousands for the fit time to be scaled quadratically. The one-vs-one scheme is used to handle the multiclass support. The hyper-parameters determine the linear kernel type to be used for this algorithm which is suitable for datasets with large features. The regularization purpose is to prevent overfitting and minimize loss function by calibrating machine learning models. Regularization is usually inversely proportional to C and has a squared l2 penalty. The probability determines the enablement of probability estimates. In the case of class_weights which have a balanced mode that uses x values in n_samples/(n_classes ∗ np.bincount(x)) as inputs, weights that are inversely proportional to the class frequencies are adjusted. The Max_iter is set to 1000 as the default, which is a solver hard limit on iterations. Finally, the data are shuffled based on probability estimates by the pseudo-random number generator.

XGBoost is derived from a gradient-boosting framework as an optimized distributed library which is portable, flexible, and efficient. It can be applied to run on billions of samples in the distributed environment with the configuration parameters as shown in Table 8. The random_state for FHE experiments is kept null. N_Estimators are the count for run numbers for the learning performed by XGBoost. The learning rate is the learning speed and the low error rate determines its proper selection. The booster parameter is used to select the model type to be run every iteration, where gbtree has tree-based models. The max_depth of a tree is specific to the learning relation of sample data for controlling overfitting at higher depth. The seed is used for parameter tuning and obtaining reproducible results. It is a supervised learning method that is focused on regression and classification with the configuration parameters as shown in Table 9. It learns simple decision rules for model prediction based on feature data. The balanced class weight means have equal weights assigned for the output class and have the same class proportion for the child samples. The max depth limits the size of the training sample nodes within the tree to avoid overfitting.

Minimum sample split is used to split an internal node based on the minimum number of specified samples. Here, the minimum sample split is related to internal nodes and the minimum sample leaf is about external nodes. The internal node has further splits but leaf nodes have no children. Minimum sample leaves are the minimum samples needed at leaf nodes. The presence of training samples in the left and right branches makes a split point be considered valid which has a smoothing effect during regression. Maximum features are used to select the number of features for the best split.

### 3.5. Detailed Comparison of ML and Concrete-ML

The x-axis presents the record count for tests that are performed by the ML algorithm on the workstation with the y-axis as time, as shown in Figure 5. It can be observed that the decision tree algorithm requires the lowest time on a workstation for testing multiple records. In the case of linear regression and SVM, the execution timing requirements are the average and are similar to each on the workstations. Whereas, XGBoost has the highest requirements on all of the prime servers. Therefore, the end user can select the appropriate algorithm based on accuracy- and time-based criteria.

Implementing ML algorithms on the AWS cloud is first presented by Figure 6. Every prime server has its respective open dataset as given in Table 3. The ML algorithm for the linear regression requires the highest time of 0.02 s and lowest of 0.001 s by DT on the prime server 1.

The highest time of 0.01 s is required by XGB and the lowest of 0.001 s by DT on the prime server 2. Whereas, the highest of 0.01 s is required by XGB and the lowest of 0.001 s by DT on the prime server 3. Overall, the DT requires the lowest processing time on all the prime servers and LR with the highest time.

In Figure 7, implementing concrete-ML on the AWS cloud has a similar behavioral pattern across the different prime servers. It can be noticed that prime server 1 performs with the lowest time, prime server 2 performs with the average time, and prime server 3 performs with the highest time for all the algorithms. Therefore, it can be concluded that cloud-ML has more algorithm-specific behavior and concrete-ML on the cloud has more pattern-based behavior by a group of algorithms.

### 3.6. Concrete-Ml Graphs Including Time (Sec) for Independent Record Encryption and Decryption

The concrete-ML algorithm time required for encryption /decryption on workstations and the AWS cloud with the VA dataset is presented in Table 10. In the case of cryptography, the encryption/decryption time for the decision tree is the lowest on the workstation, followed by XGBoost, SVC, and linear regression due to the output parameters generated. Similarly, the cloud cryptography time generated follows the same sequence, where the lowest time required is for the decision tree followed by the XGBoost, SVC, and linear regression. The processing time on the workstation is less for the concrete-ML prediction due to the higher workstation configuration as compared to the cloud given in the workstation configuration of Table 2.

### 3.7. Concrete-Ml Time for Independent Record for Multiple Datasets on AWS Cloud

Implementing multiple open heart disease datasets by concrete-ML on the cloud shows high variability in the processing with different algorithms. The Cleveland dataset has the lowest prediction time by the decision tree algorithm followed by XGB, SVC, and LR as the highest cryptographic operations, as shown in Table 11. Similarly, in the VA dataset, the lowest prediction time is given again by the DT followed by SVC, XGB, and LR. Whereas, the Hungary dataset has the lowest prediction time by the XGB followed by LR, DT, and SVC.

### 3.8. Analysis of the Final Protocol Time

The complete FHE protocol time can be given by the
FHE Protocol Total Time = Time (Cloud server communication + Encryption/Decryption + Algorithm Process)
where, Cloud Server Communication Time = Time (3 Prime Servers), and
Algorithm Execution Time = (Encryption/Decryption + Algorithm Process).

While training the prime servers with different machine learning algorithms and different open datasets, the performance details can be given as above in Table 12. The tables are summarized later in Figure 8.

The final FHE protocol is implemented on the AWS cloud network. First, the prime server is trained by the V.A. dataset and the test samples are taken from the Cleveland and Hungary datasets as shown in the Figure 8a,b. The Figure 8c,d show that the XGB requires the highest time for the FHE protocol followed by the SVC and LR which require a similar time and the least time by the DT. Similarly, when the cloud servers are trained by the Cleveland dataset, the similar pattern is again observed in the FHE protocols with different algorithms having XGB as the highest time requirements followed by the similar time of LR and SVC and lowest time requirement by the DT.

In the case of training cloud prime servers by Hungary, as shown in Figure 8c,d, XGB again requires the highest and DT the lowest time respectively. Whereas, the SVC needs more time for the execution than the LR. Overall, depending on the accuracy and time requirement, the user can choose appropriate settings for the training dataset.

The above Table 13 shows the benchmark comparison details of the multiple FHE protocol evaluations. In an outsourced multi-party k-means clustering scheme [8], multiple distinct secured keys are utilized for the protocol. This scheme proposes minimum, comparison, secure squared Euclidean, and average operations in the protocol with servers having time greater than or equal to five seconds. The multi-key homomorphic encryption (MKFHE) [26] uses the TFHE scheme with the CCS19 algorithm implemented in the cloud protocol. The MKFHE uses circuit optimization with three multi-party node protocols having preprocessing, intersection, set difference, and TH intersection for a minimum time of 5.16 s. In the case of privacy preservation in multi-layer perceptron (PFMLP) protocol [27], the improved paillier federated protocol is used that has multiple hidden layers containing multiple units with the embedded homomorphic operation. The system involves a key management center and computing server with multiple clients requiring at least 7.92 s to complete the protocol process. Federated learning has provided a key advantage to utilizing the network for healthcare applications [35,36,37]. Ultimately, it has been observed that even though the nodes involved by the benchmarking algorithms are quite similar in range to each other, the minimum processing time of 4.23 s with FHE and multiple open heart disease datasets trained evaluation are achieved by our AUTOPRO-RHC protocol.

## 4. Detail Discussion for the AutoPro-RHC Implementation

Device Availability: A hospital association with at least five to seven branches should form a contract with the wearable sensor’s device manufacturing company. Heart-affected patients staying in the remote areas should be prioritized for the device assignment and are supposed to return it after successful treatment. The AutoPro-RHC will be pre-installed before the device allotment and initiated immediately after the deployment. In the future, AutoPro-RHC can be upgraded to be easily installed on mobile devices with attached sensors for ease of usage;Data Consent and Usage: After the remote patient is diagnosed with the heart disease, the data consent form is recommended to be submitted for accepting the device agreement. Data can be optionally donated by de-identification for the hospital’s FHE model training purpose and storage. The contribution of data makes a significant difference to the scientific/medical field for the future treatment improvement purpose;Frequency of the Data Collection: Based on the patient’s location and severity type, the data can be uploaded on specific intervals. Continuous data recording is not easy to process so in such cases, an average/peak value over an interval of 10/20/30 min can be shared. In case of multi-disease category, special attention can be given by having a continuous monitoring device with emergency calling/tracking by consent;Technical Device Issues: A vendor-based maintenance system should be present to solve the device issues in case of malfunctioning or damage. Hospital-based monitoring and calling should not be responsible due to the limited resources. Effective strategy by the vendors can help to handle such problems and should be resolved on priority. Some backup sensor parts or batteries can be provided to avoid the last hour of rushing in case of an emergency situation;Future Trends and Opportunities: A mobile device can be used to collect the sensor reading, encrypt it, and share it with the AutoPro-RHC which will be more convenient. Developing a mobile application to use AutoPro-RHC will enable it to be more portable to use, carried comfortably, and charged with a regular routine. Even though the mobile application will make it more portable to use, the mobile device security must also be taken into consideration. Another opportunity is present in multiple disease diagnoses by a single device, where the patient affected with heart disease, diabetes, hypertension, etc., must be able to use a single portable device and obtain health reports weekly.

## 5. Conclusions

Heart diseases are known to be the most life-threatening condition affecting the human population. The FHE-based RHM provides a secure and effective model that provides services to the remote affected patient. The FHE model presented in this research implements a concrete-ML library within the cloud network. The prediction given by the concrete-ML is trained by using multiple open heart disease datasets. Preprocessing applied on the initial training data is to recover the missing and incomplete data and class weights are assigned. The AWS cloud FHE protocol demonstrated from this work provides the prediction from a different prime server that is grouped and reported to the patient and registered hospital. Priority schedulers help to identify the best time for the patients based on their respective conditions. Successively, the results are evaluated by the medical examiner only after the patient provides consent for results analysis. The FHE results show that the highest accuracy is achieved by support vector classification (SVC) of 88% and linear regression (LR) of 86% with the AUC of 91% and 90%, respectively. Ultimately, the best time achieved by the FHE protocol is around 4.23 s by the decision tree algorithm on the AWS cloud network. The future work will focus on utilizing the smart wearable device to analyze multiple vital signs for analyzing multiple disease-affected patients based on trustworthy computing. A mobile device or smartwatch can be used without the need to carry other sensor devices; receiving complete healthcare analysis is a necessity for this era.

## Figures and Tables

**Figure 1 sensors-23-08504-f001:**
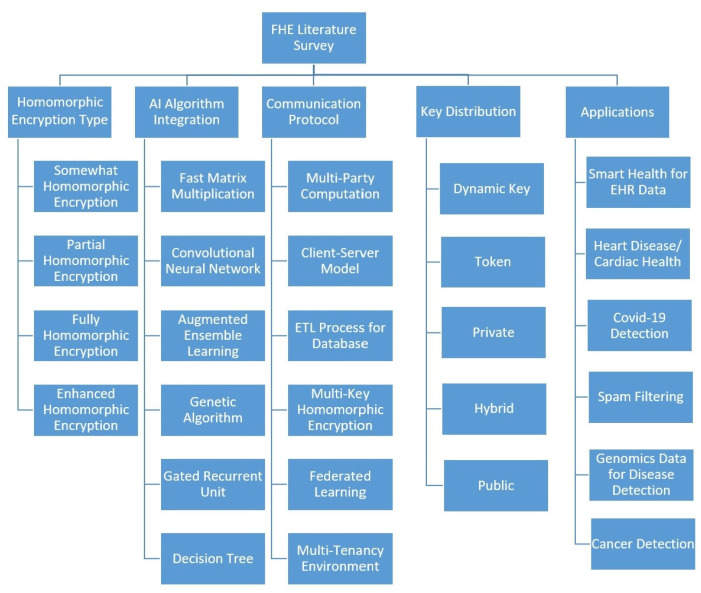
Survey for the FHE literature analysis.

**Figure 2 sensors-23-08504-f002:**
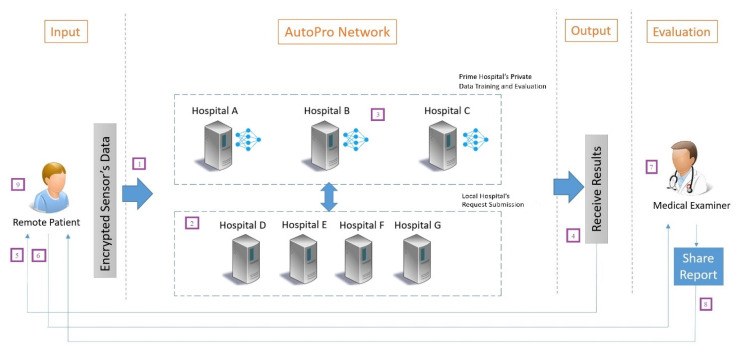
System model for the AutoPro-RHC protocol.

**Figure 3 sensors-23-08504-f003:**
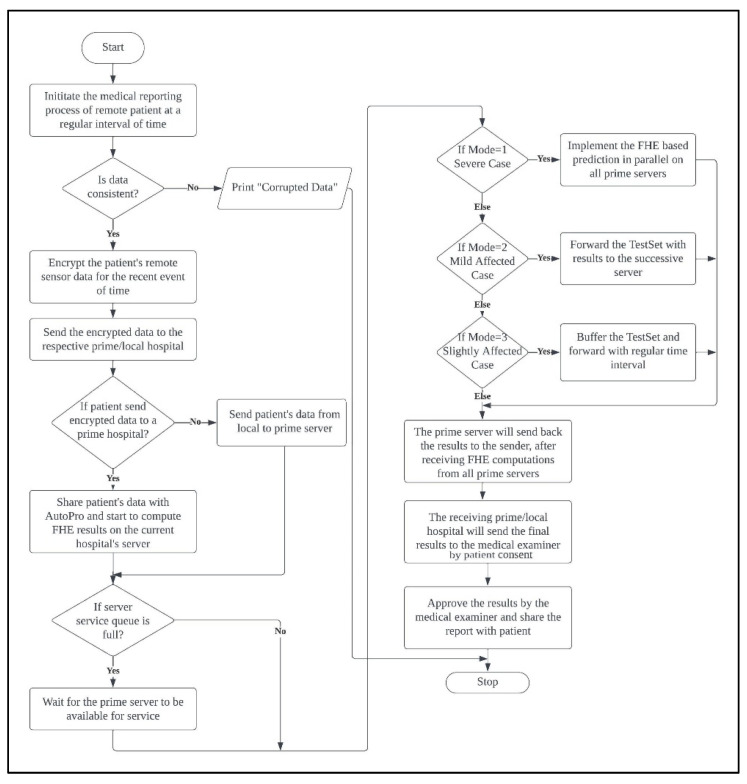
Flowchart for the AutoPro-RHC.

**Figure 4 sensors-23-08504-f004:**
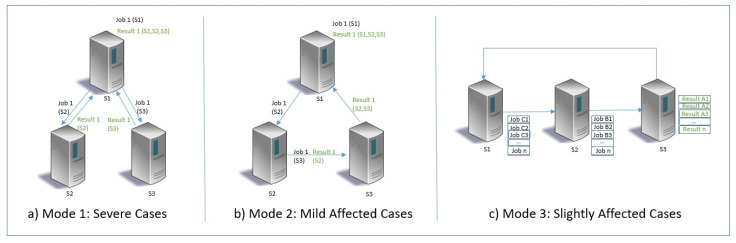
AutoPro job scheduler.

**Figure 5 sensors-23-08504-f005:**
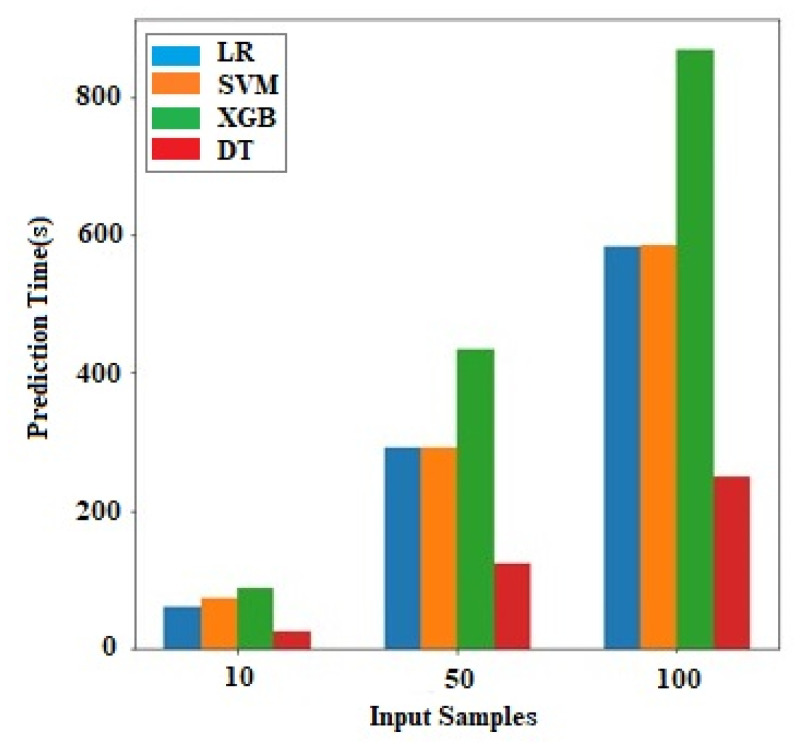
Graph comparison for the ML and concrete-ML algorithms.

**Figure 6 sensors-23-08504-f006:**
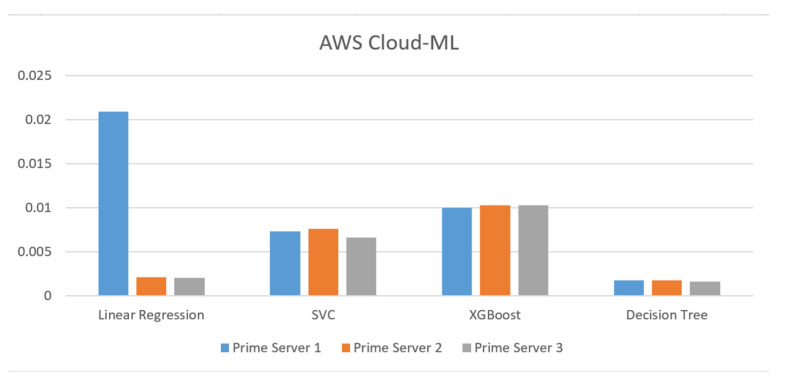
Implementation of machine learning algorithms on the AWS platform.

**Figure 7 sensors-23-08504-f007:**
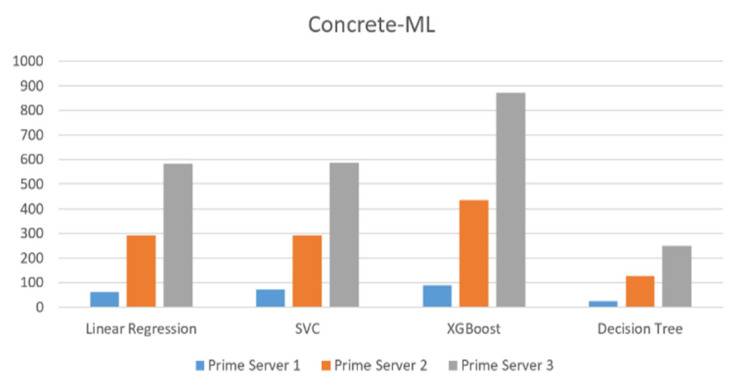
Implementation of AWS concrete-ML algorithms on AWS Platform.

**Figure 8 sensors-23-08504-f008:**
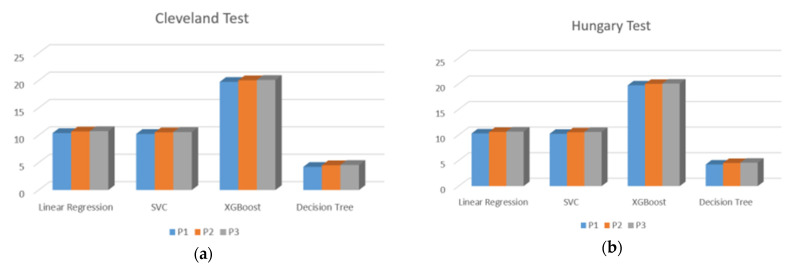
FHE protocol implementation time: (**a**,**b**) train by VA, (**c**,**d**) train by Cleveland, and (**e**,**f**) train by Hungary.

**Table 1 sensors-23-08504-t001:** Comparison of recent medical FHE algorithms.

Reference Paper	Medical Data Category	Processing Platform	Dataset	Data Pre-Processing/Protocol	FHE Algorithm	Evaluation
Zhang Li et al. (2022) [7]	Skin Lesion	Federated Learning	HAM10000 (Dermato-scopic Images)	Masking scheme and the secure multi-party computation	Dropout-tolerable scheme including DCNN	Accuracy and communication cost graphs.
Zhang P. et al. (2022) [8]	Breast Cancer Data	Cloud Computing	Wisconsin Breast Cancer Data	Secure squared Euclidean, comparison, minimum, and average protocols	Secure multi-party k-means clustering algorithm	Protocol computation and communication time, and efficiency cost graphs.
Kumari K.A. et al. (2021) [9]	Obstructive Sleep Apnea Data	Activity Tracker Device and Comput-ational Node	Fitness Tracker Dataset	Reduce data depth and cipher text errors	Gorti’s enhanced encryption scheme and Carmichael’s encryption scheme	Apnea-hypopnea index (AHI) for mild, moderate, and severe outcomes. Computation of E/D time and scheme comparison.
Shaikh M.U. et al. (2021) [10]	Arrhythmias by ECG	Cloud Computing	MIT-BIH Arrhythmia Database	QRS complex, Pan, and Tompkins algorithm	Simple FHE scheme	Sensitivity, prediction, and detection error rate.
AutoPro-RHC	Heart Disease Detection	Adaptive Autonomous Protocol (AutoPro)	UCI Heart Diseases Dataset	A novel secured multi-party computation protocol with AutoPro job scheduling	Fast fully homomorphic encryption over the Torus (TFHE) library	Recall, F1-score, accuracy, and results prediction.

E/D: Encryption/Decryption, ECG: Electrocardiogram.

**Table 2 sensors-23-08504-t002:** System configuration.

**System**	Workstation: Intel Core i7-8700K CPU@3.70GHz	Amazon Linux AMI Cloud Instance (i386, x86_64)
**Memory**	64 GB	1 GiB
**Operating System**	Ubuntu 20.04.3 LTS	Linux kernel 5.10
**Library**	Concrete.ml.sklearn, Concrete.ml.deployment, FHE ModelClient, FHE ModelDev, FHE ModelServer, Socket, OS, Sys and Time.	

**Table 3 sensors-23-08504-t003:** Dataset details.

**Dataset Name**	Heart Disease Dataset (Cleveland, Hungary and V.A.—Long Beach)
**Characteristics**	Multivariate
**Number of Instances**	303
**Attributes**	14

**Table 4 sensors-23-08504-t004:** Pre-processing Techniques and Parameter Values.

Dataset	Pre-Processing Required?	Missing Data Handling	One-Hot Encoding	Data Imbalance Handling by Class Weight
Prime Server 1 (Cleveland)	Yes	Yes	Yes	(0.92424242, 1.08928571)
Prime Server 2 (Hungary)	Yes	Yes	Yes	(0.80384615, 1.32278481)
Prime Server 3 (V.A.)	Yes	Yes	Yes	(2.1875, 0.64814815)

**Table 5 sensors-23-08504-t005:** Machine Learning based Evaluation for the Hospital Prime Servers.

Dataset	Algorithm	Parameters	Precision	Recall	F1-Score	Accuracy	AUROC	Execution Time (s)	
								Workstation	Cloud
Prime Server 1	Linear Regression	10	0.59	0.85	0.71	0.67	0.82	0.0032	0.0014
	SVM	10	0.58	**0.93**	0.70	0.67	0.82	0.0030	0.0037
	XGBoost	10	0.67	0.81	0.73	0.73	**0.83**	0.0011	0.0091
	Decision Tree	10	**0.74**	0.74	**0.74**	**0.77**	0.82	**0.0004**	**0.0012**
Prime Server 2	Linear Regression	10	0.92	0.58	0.77	0.86	0.90	0.0026	0.0014
	SVM	10	**1.00**	**0.68**	**0.81**	**0.88**	**0.91**	0.0026	0.0026
	XGBoost	10	0.92	0.58	0.71	0.82	0.84	0.0006	0.0089
	Decision Tree	10	0.85	0.58	0.69	0.80	0.83	**0.0004**	**0.0011**
Prime Server 3	Linear Regression	10	0.86	0.60	0.71	0.60	0.67	0.0017	0.0013
	SVM	10	**0.89**	0.80	**0.84**	**0.76**	0.67	0.0017	0.0017
	XGBoost	10	0.87	0.65	0.74	0.64	**0.72**	0.0006	0.0087
	Decision Tree	10	0.70	**0.88**	0.78	0.68	0.65	**0.0004**	**0.0010**

**Bold value**: Best algorithm time.

**Table 6 sensors-23-08504-t006:** FHE machine learning-based evaluation for the hospital prime servers.

Dataset	Algorithm	Parameters	Precision	Recall	F1-Score	Accuracy	AUROC	Workstation Time (s)
Prime Server 1	Linear Regression	10	0.59	0.85	0.71	0.67	0.82	345.70
	SVM	10	0.58	**0.93**	0.70	0.67	0.82	345.62
	XGBoost	10	0.67	0.81	0.73	0.73	**0.83**	521.43
	Decision Tree	10	**0.74**	0.74	**0.74**	**0.77**	0.82	**148.58**
Prime Server 2	Linear Regression	10	0.92	0.58	0.77	0.86	0.90	293.51
	SVM	10	**1.00**	**0.68**	**0.81**	**0.88**	**0.91**	291.50
	XGBoost	10	0.92	0.58	0.71	0.82	0.84	432.75
	Decision Tree	10	0.85	0.58	0.69	0.80	0.83	**125.78**
Prime Server 3	Linear Regression	10	0.86	0.60	0.71	0.60	0.67	143.82
	SVM	10	**0.89**	0.80	**0.84**	**0.76**	0.67	143.08
	XGBoost	10	0.87	0.65	0.74	0.64	**0.72**	182.01
	Decision Tree	10	0.70	**0.88**	0.78	0.68	0.65	**61.51**

**Bold value**: Best system time.

**Table 7 sensors-23-08504-t007:** SVC concrete-ML parameters.

Kernel	Regularization	C	Degree	Probability	Class-weights	Max_iter	Random-State
Linear	Yes	1	3 (default)	True	Balanced	1000 (default)	None (default)

**Table 8 sensors-23-08504-t008:** XGBoost concrete-ML parameters.

Random_State	N_Estimators	Learning-Rate	Booster	Max_Depth	Seed
0 (default)	10	0.1 (default)	GbTree (default)	3 (default)	None (default)

**Table 9 sensors-23-08504-t009:** Decision Tree Concrete-ML Parameter’s.

Class_Weight	Max_Depth	Min_Samples_Leaf	Max_Features	Min_Samples_Split
balanced	10	10	None	100

**Table 10 sensors-23-08504-t010:** Concrete-ML encryption and decryption time.

Concrete-ML Algorithm	Workstation		Cloud	
	Enc.	Dec.	Enc.	Dec.
Linear Regression	0.0047	0.0005	0.0099	0.0007
SVC	0.0047	0.0005	0.0098	0.0007
XGBoost	0.0032	0.0028	0.0054	0.0015
Decision Tree	0.0032	0.0005	0.0054	0.0007

**Table 11 sensors-23-08504-t011:** Concrete-ML Time Comparison for Multiple Datasets.

Concrete-ML Algorithm	Cleveland		VA		Hungary	
	Enc.	Dec.	Enc.	Dec.	Enc.	Dec.
Linear Regression	0.0107	0.0007	0.0133	0.0007	0.0098	0.0008
SVC	0.0097	0.0007	0.0099	0.0007	0.0111	0.0007
XGBoost	0.0053	0.0013	0.0079	0.0034	0.0061	0.0015
Decision Tree	0.0054	0.0007	0.0064	0.0007	0.0101	0.0007

**Table 12 sensors-23-08504-t012:** Train by open dataset and test on sample.

Dataset for Training	Testing Sample	Algorithm	Cloud Server Communication (sec) Time within Servers [A]			Cryptography Time (sec)		Algorithm Execution Time (sec) [D = B + C]	FHE Protocol Total Time (sec) [E = A + D]		
			M1	M2	M3	Encryption [B]	Decryption [C]		P1	P2	P3
**V.A.**	Cleveland	Linear Regression	0.95	1.24	1.30	0.0097	0.0007	9.45	10.4	10.69	10.75
		SVC	0.95	1.24	1.30	0.0095	0.0007	9.29	10.24	10.53	10.59
		XGBoost	0.95	1.24	1.30	0.0056	0.0014	18.79	19.74	20.03	20.09
		Decision Tree	0.95	1.24	1.30	0.0055	0.0007	3.28	**4.23**	**4.52**	**4.58**
	Hungary	Linear Regression	0.95	1.24	1.30	0.0095	0.0007	9.36	10.31	10.6	10.66
		SVC	0.95	1.24	1.30	0.0097	0.0007	9.28	10.23	10.52	10.58
		XGBoost	0.95	1.24	1.30	0.0056	0.0014	18.75	19.7	19.99	20.05
		Decision Tree	0.95	1.24	1.30	0.0054	0.0007	3.28	**4.23**	**4.52**	**4.58**
**Cleveland**	V.A.	Linear Regression	0.95	1.24	1.30	0.098	0.0007	9.37	10.32	10.61	10.67
		SVC	0.95	1.24	1.30	0.0105	0.0007	9.30	10.25	10.54	10.6
		XGBoost	0.95	1.24	1.30	0.0056	0.0015	22.42	23.37	23.66	23.72
		Decision Tree	0.95	1.24	1.30	0.0059	0.0007	3.79	**4.74**	**5.03**	**5.09**
	Hungary	Linear Regression	0.95	1.24	1.30	0.0110	0.0007	9.30	10.25	10.54	10.6
		SVC	0.95	1.24	1.30	0.0097	0.0007	9.35	10.3	10.59	10.65
		XGBoost	0.95	1.24	1.30	0.0061	0.0014	22.54	23.49	23.78	23.84
		Decision Tree	0.95	1.24	1.30	0.0054	0.0007	3.82	**4.77**	**5.06**	**5.12**
**Hungary**	Cleveland	Linear Regression	0.95	1.24	1.30	0.0098	0.0007	9.52	10.47	10.76	10.82
		SVC	0.95	1.24	1.30	0.0095	0.0007	11.54	12.49	12.78	12.84
		XGBoost	0.95	1.24	1.30	0.0054	0.0015	22.31	23.26	23.55	23.61
		Decision Tree	0.95	1.24	1.30	0.0056	0.0007	3.58	**4.53**	**4.82**	**4.88**
	V.A.	Linear Regression	0.95	1.24	1.30	0.0110	0.0006	9.40	10.35	10.64	10.7
		SVC	0.95	1.24	1.30	0.0097	0.0007	11.57	12.52	12.81	12.87
		XGBoost	0.95	1.24	1.30	0.0060	0.0014	21.83	22.78	23.07	23.13
		Decision Tree	0.95	1.24	1.30	0.0054	0.0007	3.58	**4.53**	**4.82**	**4.88**

**Bold value**: Best protocol time, M1: Mode 1, M2: Mode 2, M3: Mode 3, P1: Protocol 1, P2: Protocol 2, and P3: Protocol 3.

**Table 13 sensors-23-08504-t013:** Benchmark Comparison for the AutoPro-RHC.

FHE Protocol Scheme	Algorithm	Model Parameters	Time (Seconds)
Outsourced Multi-party K-means Clustering [8]	Advanced k-means clustering scheme.	Distance = 2, 3, and 4	≥5
Multi-Key Homomorphic Encryption [26]	LWE and RLWE of Tours FHE (TFHE) and CCS19.	Multi-Party Nodes = 3	5.16
Privacy Preserving in Machine Learning with HE [27]	Improved Paillier federated multi-layer perceptron algorithm	Nodes = 1, 2 and 4.	7.92
AutoPro-RHC	Concrete-ML with TFHE in the Federated Protocol	Multi-Party Nodes = 4	**4.23**

**Bold value**: Best protocol time.

## Data Availability

Not applicable.

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
