# Peer review of "Adaptive Autonomous Protocol for Secured Remote Healthcare Using Fully Homomorphic Encryption (AutoPro-RHC)"

_sensors, 2023, doi:10.3390/s23208504_

Round 1

Reviewer 1 Report

1) The algorithm of FHE is not clearly given

2) Some abbreviations are given without their terminologies. 

3) Some abbreviations are given in the first appearance without their terminologies.

1) The algorithm of FHE is not clearly given

2) Some abbreviations are given without their terminologies. 

3) Some abbreviations are given in the first appearance without their terminologies.

Author Response

Respected Sir, 

Kindly refer to the attached document for the review response. Thanks

Reviewer 2 Report

The purpose of this paper is to present the adaptive autonomous protocol on the patient’s remote health monitoring data for the hospital using fully homomorphic encryption. Some parts of the protocol are presented in the paper, but a clearer explanation of which parts of the protocol are described is missing. The originality and novelty of the work could also be highlighted more. The discussion section is missing.

In this paper are presented other medical devices used to monitor the patient's condition (lines 128-149), but the data of these devices are not used for further research. Is it necessary to include such a review in this paper?

The literature review of other authors lists the methods and tools, but does not emphasize the originality of the presented work compared to other authors.

Figure 1 provides a systematic overview of the methods, but it is very abstract and it would be appropriate to refer to the references that correspond to the descriptions given in this figure. The literature analysis does not reveal the uniqueness and originality of the presented work.

The title of Figure 2 does not reflect the content of the presented figure. The description of Figure 2 in the text is very confusing, it is not clear what actions are performed and what information is used in all steps of the communication protocol.

In line 258, it is not clear how the value ‘Mode’ is obtained and what its purpose is. No information is provided about the AutoPro job scheduler, in which part of the architecture it is implemented, or what its purpose is.

The steps of the protocol are described semiformally (steps 1-9) and a textual description is provided. It would be possible to associate steps with descriptive text.

Journal references do not provide all the detailed information (journal vol., pages, or DOI). Not all abbreviations are listed.
Figure 4 is too far from the explanatory text.
The titles of the figures and tables are not given according to the template, which complicates the analysis of the article.

The abbreviation AutoPro-RHC is used in the title of the paper, but the abbreviation SP-FHE is used most frequently in the text. It remains unclear whether this is just part of the AutoPro-RHC protocol, but clarification or unification of terms is needed.

Author Response

(The authors gave the same response as above.)

Reviewer 3 Report

The main question addressed by the research is how to design and implement an adaptive autonomous protocol for remote health monitoring (RHM) data that not only provides accurate health status reporting but also ensures the confidentiality of patients' medical data. The protocol aims to comply with the Health Insurance Portability and Accountability Act (HIPAA) guidelines, work independently within a network of hospital servers, and adapt based on the severity of patients' health conditions. The research explores the application of fully homomorphic encryption (FHE) in achieving these goals.

The topic appears to be highly relevant in the field of healthcare technology, particularly in the areas of remote health monitoring and healthcare data security. The integration of fully homomorphic encryption (FHE) to secure patient data in a remote health monitoring system is a noteworthy approach, especially given increasing concerns about data privacy and compliance with regulations like HIPAA.

The research aims to address specific gaps in the field:

  1. Security and Confidentiality: While remote health monitoring (RHM) technologies have improved healthcare access, especially for those in remote locations, concerns about the security and privacy of the transmitted data remain. The paper aims to tackle this problem by integrating FHE into the RHM system.
  2. Autonomy: Most existing systems rely on third-party services for various functionalities, but this research aims to develop a protocol that works independently within a group of prime hospital servers. This autonomy could provide more streamlined and secure healthcare services.
  3. Adaptiveness: The protocol aims to adapt based on the patient’s severity level, making it more responsive and dynamic compared to static systems. This is particularly useful for immediate and personalized healthcare services.
  4. Multiple Applications: The system is designed to be versatile, with applications mentioned like glucose monitoring for diabetes, digital blood pressure for stroke, and others, suggesting broad utility across healthcare needs.
  5. Cloud Implementation: Given that the protocol is successfully demonstrated on the AWS cloud network, this indicates a practical, scalable solution that could be integrated into existing healthcare infrastructures.

the research appears to make several significant contributions to the subject area of remote health monitoring and healthcare data security compared to other published material:

  1. Fully Homomorphic Encryption (FHE) in Healthcare: While encryption methods have been applied in healthcare, the use of FHE in remote health monitoring is less common. FHE allows computations on encrypted data without needing to decrypt it, offering a robust security measure for confidential patient data.
  2. Adaptive Autonomous Protocol: The paper proposes an autonomous protocol that operates within the hospital servers, removing dependency on third-party systems. This can potentially increase the integrity and confidentiality of the data, while also allowing for quicker decision-making.
  3. Severity-based Adaptiveness: Unlike many static systems, the proposed protocol adapts based on the severity of the patient's condition ("slight, medium and severe cases"). This can allow for more targeted and immediate healthcare interventions, making the system more dynamic and responsive.
  4. Broad Range of Applications: The system isn't limited to a single application but can be applied to multiple health monitoring scenarios like glucose levels, blood pressure, pulse oximetry, etc. This multi-application approach increases the system’s utility and can help streamline various aspects of remote healthcare delivery.
  5. Open Dataset and Reproducibility: Using an open dataset for training the machine learning model ensures that other researchers can validate or build upon this work, strengthening its academic value.

Specific Improvements:

  1. Validity and Generalizability: Since the research uses an open dataset for heart diseases, the authors could consider applying their protocol to other datasets as well, to ensure the model is generalizable across multiple conditions.
  2. Encryption Overhead: Fully Homomorphic Encryption is computationally expensive. The authors should detail how much computational time and resources are required for the FHE implementation, and how that would scale in a real-world healthcare scenario.
  3. User Experience: Since one of the goals is to save patient time, user experience for both healthcare providers and patients should be considered. Is the system user-friendly, and how does the encryption process affect user interaction with the system?
  4. Security Analysis: A detailed security analysis should be conducted to assess any potential vulnerabilities in the autonomous protocol and FHE implementation, possibly with penetration testing.
  5. Ethical Considerations: With healthcare data, ethical considerations are paramount. Do the authors have a method for obtaining informed consent if deploying this in real-world scenarios?
  6. Long-Term Sustainability: How does the system perform over a long period? Is it stable and robust enough to handle the fluctuations in data and potential system errors or attacks?

Further Controls:

  1. Redundancy Checks: Incorporate measures to validate the integrity of the data before and after encryption/decryption.
  2. Randomized Control Trials: If possible, implementing a real-world trial in a controlled environment could provide rigorous evidence for the system's efficacy.

 Moreover:

1) Authors should explicitly declare the limitations of the study

2) The number of references to related works could still be increased (no references from 2023)

3) More discussion of possible future works could be added in the conclusions

Minor editing of English language required are required

Author Response

(The authors gave the same response as above.)

Round 2

Reviewer 2 Report

The authors improved the article after revision. A few text formatting errors remain to match the template provided by the journal.

Author Response

Respected Sir,

We have updated the manuscript based on your recommendations for the sensor's template format update as follows:

  1. The font type and size are made consistent across all of the manuscript.
  2. Section and subsection numbering is updated.
  3. Discussion section as renumbered to Section 4.
  4. In experiments section, formula discussion and its fonts are updated.
  5. Graph comments are updated.
